# Health Functionality and Quality Control of Laver (*Porphyra*, *Pyropia*): Current Issues and Future Perspectives as an Edible Seaweed

**DOI:** 10.3390/md18010014

**Published:** 2019-12-23

**Authors:** Tae Jin Cho, Min Suk Rhee

**Affiliations:** Department of Biotechnology, College of Life Sciences and Biotechnology, Korea University, 145, Anam-ro, Seongbuk-gu, Seoul 02841, Korea; chshatria@korea.ac.kr

**Keywords:** raw laver, processed laver product, edible seaweed, nutritional value, functional substance, health functionality, processing technology, microbial risk, chemical risk, omics-based technology

## Abstract

The growing interest in laver as a food product and as a source of substances beneficial to health has led to global consumer demand for laver produced in a limited area of northeastern Asia. Here we review research into the benefits of laver consumption and discuss future perspectives on the improvement of laver product quality. Variation in nutritional/functional values among product types (raw and processed (dried, roasted, or seasoned) laver) makes product-specific nutritional analysis a prerequisite for accurate prediction of health benefits. The effects of drying, roasting, and seasoning on the contents of both beneficial and harmful substances highlight the importance of managing laver processing conditions. Most research into health benefits has focused on substances present at high concentrations in laver (porphyran, Vitamin B_12_, taurine), with assessment of the expected effects of laver consumption. Mitigation of chemical/microbiological risks and the adoption of novel technologies to exploit under-reported biochemical characteristics of lavers are suggested as key strategies for the further improvement of laver product quality. Comprehensive analysis of the literature regarding laver as a food product and as a source of biomedical compounds highlights the possibilities and challenges for application of laver products.

## 1. Introduction

Lavers are red seaweed species mainly consumed as processed food products or used as a source of health-promoting substances. They belong to the genera *Porphyra* and *Pyropia* (which contains many species formerly included in *Porphyra*) (Phylum: Rhodophyta; Class: Bangiophyceae; Order: Bangiales; Family: Bangiaceae) [1]. Traditionally, lavers were staple foods in limited regions of Asia, but increased awareness of their health benefits and the globalization of processed food products has led to dramatic increases in consumption across the world [2]. The growth of global seaweed aquaculture as a source of pharmaceuticals and biomaterials (e.g., Alga Technologies, Cyanotech, etc.) is expected to contribute to the expansion of the laver industry [3,4]. Global laver production has increased from 517,739 t/US $945.1 billion in 1987 to 841,131 t/US $1285.0 billion in 1997; 1,510,911 t/US $1403.9 billion in 2007; and 2,563,048 t/US $2319.7 billion in 2017 (Figure 1) [5]. Commercial production of laver products (e.g., gim snack (seasoned laver), mareun-gim (dried laver), okazu nori (laver for side dish), yakinori (roasted laver), zicai tang (laver soup), etc.) are concentrated in northeastern Asia, with South Korea, China, and Japan producing 99.87% of total world production in 2017 (Figure 1) [5]. This reflects the traditional consumption of laver in these countries and also regional environmental conditions favorable for aquaculture [6].

Currently, Algaebase [7] lists 188 and 77 species for *Porphyra* and *Pyropia*, respectively. As shown in Table 1, species found mainly in northeastern Asia and suitable for aquaculture have been used as the target organisms for research studies on the food production (raw and/or processed laver) and/or substances potentially beneficial to health (e.g., porphyran, taurine, vitamins, etc.). Previous studies on laver exploitation focused on the main species produced commercially in the Republic of Korea (*Pyropia tenera* (Kjellman) Kikuchi et al., 2011, *P. yezoensis* (Ueda) Hwang and Choi, 2011, *P. seriata* (Kjellman) Kikuchi and Miyata, 2011, *P. dentata* (Kjellman) Kikuchi and Miyata, 2011), China (*Pyropia haitanensis* (Chang and Zheng) Kikuchi and Miyata, 2011, *P. yezoensis*) and Japan (*Pyropia tenera, P. yezoensis, P. pseudolinearis* (Ueda) Kikuchi et al., 2011) [8].

Laver can be consumed as food, either raw or processed (e.g., dried, roasted, seasoned) or as a source of substances beneficial to health. In northeastern Asia, laver is consumed mainly as a side-dish and thus is generally perceived as a foodstuff rather than as a source of health functionality substances. By contrast, studies of other edible seaweeds (e.g., green or brown algae) focused mainly on their non-food roles as sources of nutraceuticals, food additives, and biomaterials. Nutritional values and bioactive components of algal species linked to major health benefits were reported, highlighting the potential for growth of the laver industry as both an edible seaweed and a source of useful compounds [4,9,10,11,12,13,14,15,16,17,18]. However, although practical studies of the utility of laver as an edible seaweed reported the distinct characteristics of various product types (i.e., material composition, effects of manufacturing processes), there is no comprehensive analysis of the literature regarding the nutritional/functional characteristics of laver and the technological basis for its quality control.

This review evaluates the results of research into the nutritional/functional characteristics of laver products (consumed as food or for health benefits) and the application of technology to those products, through the categorization of current issues (Section 2 and Section 3) and discussion of future perspectives (Section 4). To analyze advances over the decade 2009 to 2019, we review literature retrieved from the following databases: PubMed, EBSCO, Research Information Sharing Service (RISS), National Digital Science Library (NDSL), SCOPUS, Web of Science, and WIPO IP Portal. The aims of this work are (1) to comprehensively review recent findings on the utility of edible lavers in both raw and processed products and (2) to identify priority areas for future research on the exploitation of lavers.

## 2. Food Products Containing Lavers

Food products may contain either raw or processed lavers. Studies reporting the distinct nutritional/functional values for each product type are reviewed here, the major factors determining the expected effects of consumption are identified, and their implications are discussed. Whereas it should be noted that since overall dietary habit defines total intake of nutritional substances with potential health functionality, consuming those substances is likely to have no effect on health (if total intake is already adequate) or even results in negative health consequences (in case of excessive intake).

### 2.1. Raw Lavers

Published analyses of the nutritional and health benefits of raw laver can be divided into studies of raw wet laver directly consumed as edible seaweed and studies of raw laver pre-treated for measurement of dry weight composition. Nutritional values vary widely among product types due to the high-water content of raw wet laver (Table 2). The water content of raw wet laver was generally reported as ca. 90% [19] (Table 2). Thus, data on compounds of nutritional and health value available in raw wet laver can only be estimated by analysis of raw wet laver itself rather than the dehydrated product used for dry weight-based analysis. However, most previous studies of nutritional values presented analytical data based on dry weight [20,21,22,23,24,25,26,27], and wet-weight values are rarely reported [19]. Since seasoned raw wet laver is often consumed without dehydration or further processing, data on nutritional value relative to wet weight are needed to accurately estimate the potential health benefits.

With respect to the nutritional composition of raw laver by dry weight (Table 3), most parameter values are similar to those obtained from other edible seaweeds (carbohydrate, dietary fiber, protein, lipid, and ash). Other nutritional components analyzed for raw laver are those generally used for assessing the health benefits of edible seaweeds, namely minerals, fatty acids, amino acids, sterol, antioxidants, phenolic compounds, pigments, and vitamins.

### 2.2. Processed Laver Products

The CODEX regional standard for laver products (CXS 323R-2017) categorizes these according to the processing methods (i.e., dried, roasted, seasoned) and raw materials (i.e., single or multiple edible seaweed species) [28]. Dried laver is the most common type and can be divided into primary dried products (i.e., washed, chopped/cut, molded, dehydrated, and dried after harvesting) and secondary dried products (made by re-drying primary dried products for long-term storage). Roasted laver is dried laver roasted without seasoning, while seasoned laver is dried material treated with a variety of ingredients using several processing methods (e.g., roasting, frying, treating with edible oil) before or after seasoning. The seasoned category also includes laver seasoned for brewing and broken and roasted/stir-fried dried laver seasoned for consumption after addition of boiling water. Maximum water contents of primary dried, secondary dried, roasted, and seasoned laver are set as 14%, 7%, 5%, and 5%, respectively. Nutritional values of processed laver products show the expected range of composition as described for raw laver in Section 2.1 (Table 3). Moreover, as noted for each product type, this variation in the nutritional/functional content implies a need for further analysis of nutritional values of specific product types to accurately estimate the potential for health benefits arising from the consumption of those products. Composition of processed laver products is standardized according to the CODEX regional standard (CXS 323R-2017) for products based on *Pyropia* spp. (and containing other optional ingredients) [28]. Edible seaweed other than *Pyropia* spp. is one of the major optional ingredients and can be intentionally or unintentionally included in processed laver products. Multiple species of edible seaweeds such as *Ulva* spp. (green laver), *Ecklonia cava* Kjellman, 1885, or *Capsosiphon fulvescens* (C. Agardh) Setchell and Gardner, 1920, can be combined in a single processed laver product to improve palatability or to create specific organoleptic characteristics. These optional ingredients have distinct nutritional and potential for health-promoting features [29]. The nutritional value of the combined product depends on raw material composition and enhancement of product quality with respect to potential health functionalities and organoleptic properties [30].

Nutritional values obtained from processed laver products (i.e., dried, roasted, or seasoned) are broadly similar (Table 3). However, major differences in nutritional values and product quality can arise as a result of processing method, either by drying of raw laver or subsequent roasting or by seasoning. The drying process can affect various nutraceutical components including dietary fiber, phenolic compounds, pigments, and antioxidants [36]. The higher content and bioavailability of Vitamin B_12_ (VitB_12_) in raw, as opposed to dried, laver may imply the conversion of VitB_12_ in raw laver to its analogues (which are not bioavailable to mammals) by the air-drying process [37]. Drying of laver by lyophilization was suggested to prevent loss of bioactive VitB_12_ [38]. With respect to further processing, roasting or deep-frying of dried laver results in decreased mineral content (calcium, iron, magnesium, phosphate, potassium). Deep-frying (160–180 °C for 10 sec) decreased mineral content by a factor of 2–6 compared with roasting (300 W for 2 min) [39]. Simulated domestic cooking of dried, roasted, and seasoned laver showed consistent trends in (1) decreased water content (9.69%, 3.66%, and 1.49% for dried, roasted, and seasoned laver, respectively), (2) denaturation of amino acids (mainly glycine, citrulline, valine, isoleucine, leucine, and gamma-aminobutyric acid), and (3) higher mineral contents in dried than in roasted or seasoned laver (calcium: dried 4976 mg/g, roasted 2202 mg/g, seasoned 2037 mg/g; potassium: dried 31210 mg/g, roasted 29540 mg/g, seasoned 28800 mg/g; zinc: dried 45.12 mg/g, roasted 24.33 mg/g, seasoned 18.37 mg/g; copper: dried 6.49 mg/g, roasted 6.11 mg/g, seasoned 4.71 mg/g) [40]. However, lower contents of functional substances in processed laver products than in dried laver does not always implicate the processing steps as causal factors: Although VitB_12_ content was lower in seasoned (51.7 μg/100 g) than in dried laver (133.8 μg/100 g), the destruction of VitB_12_ by the roasting process was not detected, and thus, the addition of optional ingredients (e.g., seasoning) is thought to be the cause of the difference [41]. With respect to other measures of product quality, roasting may cause color deterioration by its effects on pigments such as chlorophyll [42]. Since the processing of laver has acted as essential role for the quality control of products (e.g., drying greatly improves shelf-life of laver to facilitate wider accessibility for individuals as a food source), the application of optimal processing conditions ensuring both nutritional/functional values and the product quality is important.

Lavers can be a component of other food products such as seaweed chips, salads, laverbread, laver cake, onigiri (rice ball), kimbab (seasoned rice roll), and laver soup [10,13,15,16]. However, previous studies of these processed products focused on organoleptic characteristics with sensory evaluation, rather than on potential for health and/or nutritional benefits.

## 3. Lavers as Functional Foods: Unique Health Benefits of Laver

Seaweeds have long been regarded as a rich source of health-promoting substances. However, relevant studies mainly focused on internationally produced and consumed seaweeds, including brown [43] and green algae [44], and on red algae other than laver [45]. This section reviews the available information on the health benefits of lavers, with particular reference to active substances unique to this group of edible seaweeds. Table 4 summarizes the major components of laver linked to well-known health benefits based on the currently reported research studies investigating the putative health effects of each component, highlighting the necessity for future research regarding long-term outcomes of laver consumption on human health.

Since most of the reported health benefits of laver could also be gained by consumption of other edible seaweeds [11], this section focuses on studies that report unique health functionalities of laver derived from components not found in other edible seaweeds, namely, porphyran, VitB_12_ and taurine [9,10].

Porphyran is the distinctive dietary fiber found in laver, and its health effects were intensively studied to determine the nutritional/functional quality of lavers [83]. Important bioactivities that can be attributed to porphyran include anti-cancer, antioxidant, and anti-inflammatory effects and/or immunomodulation and prevention of diseases such as cardiovascular, nervous, bone, and diabetic disorders [9,10]. Those bioactivities have been demonstrated by the examination of porphyran extracted from laver species as reported in following research studies. The anti-cancer effect of porphyran was demonstrated by using human cell lines including the hepatic carcinoma (Hep3B), cervical cancer (HeLa), and human breast carcinoma (MDA-MB-231) cell lines [53]. The anti-cancer effect of porphyran was also evaluated using human gastric cancer cells through the induction of apoptosis [46] and the inhibition of cell proliferation [47]. In dietary experiments using rats, prevention of cardiovascular disease may be achieved by the anti-hyperlipidemic effect revealed by the decrease in serum cholesterol level [57,84]. The basis of these effects was shown to be reduced secretion of the essential component of very low-density lipoprotein (VLDL) in blood (i.e., apolipoprotein B100) [58]. Properties of porphyran derivatives were also reported, especially for antioxidant effects assessed by radical scavenging and reducing power [62,63]. Immunomodulatory effects were shown as immune responses to myelosuppression by the oral administration of porphyran to rats [68]. Anti-inflammatory activity was evaluated by inhibition of secretion of inflammatory markers (nitric oxide and tumor necrosis factor alpha) by macrophages (RAW264.7 cell) [72], and the suppression of activation of immune cells [73]. Neuroprotective effects may be attributable to oligo-porphyran, with the mechanism for the protection of neurons shown to be regulatory effects linked to the inhibition of apoptosis in neuronal cells [77]. Therapeutic effects against bone diseases by suppression of osteoclast formation induced by the receptor activator of nuclear factor κB ligand were demonstrated using RAW 264.7 cells [78].

The beneficial health effects of the vitamin B complex (e.g., choline, inositol) in laver include the synthesis of major nutritional factors (i.e., carbohydrate, protein, and lipid), anti-cancer effects, and enhanced immunomodulation [18,56,85,86]. Lavers produce exceptional quantities of VitB_12_ and thus can be used to counter the deficiency of VitB_12_ (e.g., methylmalonic acidemias) in vegan diets by the consumption of laver [86,87,88]. The bioavailability of VitB_12_ was also confirmed by increases in the hepatic VitB_12_ level of rats by the intake of laver [38,89] and by the release of VitB_12_ from laver after human consumption simulated through in vitro gastrointestinal digestion experiments [41].

Taurine is a major amino acid in laver and other red algae but rarely present in brown or green algae [90]. Decreased plasma cholesterol levels in rats after the consumption of taurine were reported [56,61]. Promotion of neuronal development by taurine in laver extracts was also experimentally demonstrated by the primary culture of hippocampal neurons [76].

## 4. Future Perspectives on Technical Advances in Laver Utilization

The main current issue in laver utilization is the elimination of potential risk factors in the process from farming to the manufacture of laver products. For the future, we need to consider the application of new technologies for the identification of useful constituents. This section covers the development of technologies for the control of chemical/microbiological risks and novel techniques that may promote the consumption of lavers as edible seaweeds, in particular, omics-based research linked to the health benefits of lavers.

### 4.1. Management of Current Issues: Control of Potential Risks from Farming to Processing of Lavers

The sequence from cultivation of laver in aquaculture farms to the final drying steps [91] is common to most processed laver products because roasted and seasoned laver are manufactured using dried laver as a basic material.

For processed laver food products, effective control of chemical and microbiological risk factors is the prerequisite for human consumption. Since major risk factors and their extent vary across the production process, it is important to establish intervention strategies based on a detailed understanding of the overall production process. Management strategies effective in the identification and control of chemical/microbiological risks should also be established for individual product types.

#### 4.1.1. Control of Chemical Risks

The chemical risks can be defined as the consumption of excessive levels of substances that lead to side-effects and/or the exposure to toxic agents (such as heavy metals) in laver products [92,93]. Major food constituents with side-effects are iodine, fibers, and sodium (in seasoned laver). Iodine has beneficial effects on thyroid gland functioning, but excessive intake should be avoided to prevent potential adverse effects such as autoimmune thyroiditis or hypothyroidism [94,95,96,97,98]. Overconsumption of fibers can lead to vomiting or abdominal pain with diarrhea in people with sensitive stomachs and may cause dyspepsia even in healthy people due to generation of gases in the digestive system [99]. The much greater sodium content of seasoned (relative to dried) laver shows that salts added during processing can result in consumption of excessive levels of sodium [32]. Heavy metals (As, Cd, Cr, Cu, Hg, Ni, Pb, Zn, etc.) were detected in both raw and processed laver products (Table 5). It should be noted that the contents of heavy metals in laver have been reported as variable according to a range of factors including the cultivar, species, season, and processing conditions [100,101]. Most previous studies recorded acceptable levels of contaminants according to the hazard quotient (HQ) of heavy metals in laver products or provisional tolerable weekly intake (PTWI) set by the Food and Agriculture Organization/World Health Organization (FAO/WHO) [42,102]. Guidance values for tolerable intake [PTWI, provisional tolerable monthly intake (PTMI), provisional maximum tolerable daily intake (PMTDI)] for major heavy metals detected from laver were set as follows: Al (PTWI 2.0 mg/kg bw/week), Cd (PTMI 25 µg/kg bw/month), Cu (PMTDI 0.5 mg/kg bw/day), and Hg (PTWI 4.0 µg/kg bw/week) [103]. However, high levels of aluminum in laver (388.6–623.4 mg/kg dry weight) were reported as indicators of food pollution [104]. In addition, arsenic is the major heavy metal contaminant in laver [42,102,105,106], and the potential risk of production of toxic metabolites by the human digestive process was also stressed [107,108].

Setting recommended-intake limits to prevent overconsumption of nutritional components (i.e., iodine, dietary fiber, sodium) that pose potential chemical risks is the primary strategy for risk management, and specific control methods for these factors are generally not required. By contrast, since reducing the heavy metal content of laver can lower the risks, intervention technologies for the elimination of heavy metal contaminants were developed. Cadmium, chromium, and lead can be removed by immersion of laver in acid solution (citric, hydrochloric, or nitric) of pH 2.5–4.0 for 20 min, and this method could be applied because laver undergoes color changes only in more acidic conditions (pH 2.0) [111]. Heavy metal contents of processed (roasted or seasoned) laver products indicate a reduction in the levels of lead, mercury, and cadmium during the cooking process [40]. However, the increase in bioaccessible arsenic after human digestion may be a result of the roasting process [10,112]. A correlation between the arsenic content of laver and that of seawater in the cultivation area was also reported, suggesting environmental management as one of the risk control strategies [105].

#### 4.1.2. Control of Microbiological Risks

Potential microbiological risks can be identified in the national standards and regulations for laver products. China, in particular, has strict regulations (GB 2733-2005) specifying maximum permitted values for aerobic plate counts (APC; 30,000 CFU/g), coliforms (30 MPN/100 g), mold (300 CFU/g), *Salmonella* spp. (not detected), *Vibrio parahaemolyticus* (Fujino et al.) Sakazaki et al., 1963 (not detected), *Staphylococcus aureus* Rosenbach, 1884 (not detected), and *Shigella* spp. (not detected) [113]. However, laver is generally contaminated with marine bacteria [114,115], and inappropriate processing conditions may allow the growth of contaminants not only from the natural habitat but also from surrounding environments [42,116]. The importance of microbiological quality was stressed by contamination reports relating to raw and processed laver [117,118] and also to food products based on laver (e.g., sushi, kimbab) [119,120].

Table 6 summarizes microbiological contamination data for laver categorized as (1) processed laver products, (2) food products containing laver, and (3) intermediate and end-products from manufacturing plants. Aerobic plate count is the main indicator of microbiological quality. An APC value of 6.5 log CFU/g recorded from commercial dried laver products [121,122] indicates a high level of contamination. Microbiological quality factors represented by viable cell counts and coliforms were also reported for dried (7.6 log CFU/g and 3.2 MPN/100 g, respectively) and roasted laver (7.5 log CFU/g and 3.7 MPN/100 g, respectively) as the end-products from manufacturing plants. The close similarity of these values suggests that the processing of dried into roasted laver has a very limited antimicrobial effect [123,124]. Microbial populations of commercial products also differ according to product type (APC: 6.9, 3.4, and 4.9 log CFU/g; coliforms: 2.1, 1.6, and 1.0 log CFU/g, for dried, roasted, and seasoned laver, respectively) [125]. Aerobic plate count, coliforms, yeast/mold, and *Bacillus cereus* Frankland and Frankland, 1887 from processed laver products were also reported as 4.3–7.2 log CFU/g, 1.9–2.2 log CFU/g, 2.1–4.9 log CFU/g, and 2.3 log CFU/g, respectively [126]. Microbiological quality factors (APC, coliforms) were used to identify laver as the main source of microbial contaminants in ready-to-eat foods (e.g., kimbab) containing this edible seaweed: dried laver APC: 8.8 log CFU/g [127,128,129]; dried and roasted laver APC: 6.0–7.0 log CFU/g; and coliforms: 2.0–3.0 log CFU/g [130]. *Bacillus cereus* (detection rate: 12%) and *Clostridium perfringens* (Veillon and Zuber) Hauduroy et al. 1937 (detection rate: 3%) were also reported in dried laver [131]. In the case of intermediate and end-products from the manufacture of processed laver products, changes in the microbial level for each step indicate critical control points for the management of microbiological risks. Intermediate products from the manufacture of dried laver (i.e., after primary scrubbing in salt water, primary debris elimination, secondary scrubbing in salt water, secondary debris elimination, chopping and scrubbing in fresh water, molding, drying, and packaging) from seven companies showed an increase in total viable cell count (TVC) during the drying step (final products TVC: 5.6–8.0 log CFU/g; coliforms: 54–27,600 MPN/100 g) compared with the first manufacturing step (primary scrubbing in salt water TVC: 1.5–2.8 log CFU/g; coliforms: 18–75 MPN/100 g). This indicates the drying step as a critical control point for the microbiological quality of dried laver [42]. Samples collected from six companies producing seasoned laver showed high levels of microorganisms in the dried laver raw material (APC: 4.4–7.8 log CFU/g; coliforms: 54–27,600 MPN/100 g). Changes in microbial counts during the manufacturing process (i.e., primary roasting, seasoning, secondary roasting, counting, and packaging) highlighted second roasting as the key intervention step for microbial control [116]. Sequential changes in APC at each stage in the manufacture of seasoned laver (i.e., primary roasting, secondary roasting, counting, and packaging) also indicate secondary roasting as the most effective decontamination process [132,133].

Antimicrobial treatments to mitigate potential risks were applied to a range of product types from raw materials to processed foods (Table 7). Little research was conducted into decontamination of harvested raw laver because subsequent manufacturing steps (washing, drying, roasting) have generally been considered effective for the control of microbial contaminants. Park et al. [134] reported that exposing *Bacillus cereus* and *Escherichia coli* (Migula) Castellani and Chalmers, 1919 in raw laver to 200 ppm NaOCl with 60 min of ultrasound could achieve reductions of 2.6 and 3.2 log CFU/g, respectively. Decontamination methods can be applied to both processed laver products and to other processed foods containing lavers. Gamma irradiation (3 kGy for 24 h) used as an antibacterial treatment for kimbab [120,127,130] and dried laver [127] produced a reduction of up to 2 log in mesophilic bacteria (initial population: 6.0–8.8 log CFU/g), and an inactivation of foodborne pathogens to levels undetectable by plate-count methods (e.g., *Salmonella typhimurium*, *Staphylococcus aureus*, and *Listeria monocytogenes* (E. Murray et al.) Pirie, 1940; initial population: 6–7 log CFU/g). Corona discharge plasma jet and low-pressure air plasma achieved a 1.5 log reduction in mesophilic bacteria on dried laver without post-treatment color changes [135]. An electron beam (e-beam) can be used as a more energy-efficient alternative to gamma irradiation. A 4 kGy treatment achieved a 1.4 log CFU/g reduction of APC from dried laver [136]. Using heat-assisted low dose e-beam irradiation, coliforms could be eliminated from dried laver (> 1.5 log reduction) by 1–4 kGy without any changes in color or pigment contents [137,138]. Optimal treatment conditions ensuring product quality (i.e., thermophilic acidophilic bacterial count: ~10^3^ CFU/g; water content: 5%; acceptable palatability score from sensory evaluation) were suggested by response surface analysis as an irradiation dose of 1.8–3.0 kGy and heating at 154–170 °C for 10–18 s [139]. Not only for research studies regarding the development and application of intervention methods for microorganisms, patents specified for laver processing have been reported including raw wet laver [140], dried laver [141,142], and roasted laver [143]. However, although decontamination technologies reported from both academic research cases and patents were shown to reduce microbial populations to acceptable levels, there is major limitation in burden on the adoption of additional treatment devices for manufacturers. Since most those techniques are physical treatments generally applied to final products, further research into chemical/physicochemical treatment technologies applicable from intermediate products to end-products is required to support the wider application of microbial risk management strategies by manufacturers.

### 4.2. Future Issues: Identifying the Health-Promoting Properties of Laver

Omics-technology is the most advanced research method for detailed understanding of the biological characteristics of edible seaweeds. This section covers the recent findings from the omics-technologies applied to laver as a future issue for obtaining useful information linked to the health functionalities and quality control of laver including the growth characteristics and biochemical composition. Especially since these research studies have also identified the key determinant factors on the quality of laver with the perspectives to its potential health-promoting properties, the improvement on long-term outcomes of laver consumption on human health can be expected, and thus, practical evaluation should be followed as further studies. The genome, microbiome, transcriptome, proteome, and metabolome of laver species were all analyzed (Table 8).

The plastid and mitochondrial genomes of red algae were first analyzed in *Porphyra purpurea* (Roth) C. Agardh, 1824 [144,145,146], with subsequent genomic research conducted using next-generation sequencing technologies [146,147,148,149,150,151,152,153,154]. Laver is regarded as the model genome for red algae, and detailed analysis has provided data for phylogenetic, taxonomic, and evolutionary studies [146,147,155]. Genomic data from *Porphyra* and *Pyropia* species were generated by whole genome sequencing (WGS) and comparative genomics (Table 8). WGS of laver has allowed identification of genomic features associated with nutritional/functional characteristics [147] and/or product quality (e.g., color) [154]. In *Porphyra umbilicalis* Kützing, 1843, structural characteristics are linked to major nutritional/functional constituents and are believed to confer resistance to stressful habitat conditions, such as repeated desiccation and rehydration in the intertidal zone [147]. WGS of *Pyropia yezoensis* followed by annotation of major functional genes governing photosynthesis has identified new genes involved in the control of laver color. Designation of biomarkers based on these gene sequences is expected to provide practical information to the laver industry, allowing prediction of color and also details of diseases responsible for color fading [154]. Genomic data reveal the sequences for gene sets associated with the specific metabolism of laver, and this genome-wide identification can be used to target key functional genes for further transcriptomic analysis [156]. Comparative genomic analysis is an effective method of species determination in morphologically simple organisms such as lavers. It was suggested that destructive sampling and DNA extraction from fragmentary material may be useful in the identification of type specimens [146]. Phylogenetic analysis can clarify the distinct characteristics of laver compared with other red algae [153]. Complete mitochondrial genome analysis can also allow recognition of unidentified algal species by comparison of gene sequences with those of morphologically similar organisms [148]. Phylogenetic analyses of *Pyropia* and *Porphyra* also broaden our understanding of biodiversity within these genera, with *Pyropia haitanensis* [150] and *P. yezoensis* [151] providing examples of this. Biomarkers for the differentiation of laver cultivated in different regions can be identified from gene sequences based on the divergence in genomic features [152].

From the perspective of the laver industry, transcriptomic analysis can provide a detailed understanding of the unique life cycle (linked to laver yields) and stress response (linked to laver quality) (Table 8). Advances in applied transcriptome analysis include identification of housekeeping genes involved in internal control processes [157]. With respect to the life cycle, previous studies focused on characterization of the evolutionary aspects of the distinct stages [158,159], development of applied technologies including the designation of biomarkers [160], and treatments affecting the regulation of gene expression linked to reproductive processes [161]. A major focus for genomic research has been the interpretation and/or comparative analysis of transcriptomic data relating to the responses of laver species to abiotic stress [162,163], principally environmental conditions such as high temperature [164,165,166], repeated desiccation-rehydration [167,168], and hypersalinity [169].

The microbiota of laver is generally host-specific and changes according to the physiological state of the alga or according to factors in the surrounding environment [170]. Because the spatial heterogeneity of microbiota on laver can make accurate characterization of microbial communities difficult, a database of the determining factors affecting microbiome analysis was established [171]. Changes in the microbiome can be regarded as indicators of environmental conditions in the farming area [172,173]. Analysis of the microbial community of laver has revealed not only seasonal variation in the microbiota but also evidence of host-microbiota interactions from the identification of bacteria affecting the morphogenesis and growth of laver [155]. In the sea water of seedling pools, the emergence of filament diseases resulted in clear changes in microbial community composition. Healthy seedling pools showed a dominance of *Sediminicola* spp. and *Roseivirga* spp. Seedling pools with diseased laver showed increased abundance of *Vibrio* spp. and *Polaribacter* spp., suggesting they may be suitable biomarker organisms for monitoring laver disease [174].

Proteomic studies of the impact of stress factors on laver have focused on stress-response mechanisms and the effects of mutation. Abiotic and biotic stresses associated with the sequential steps from farming to processing of laver include high temperature [175,176], desiccation [177,178], and pathogen infection [179]. High temperature is one of the main determining factors of nutritional/functional quality and yield for laver strains able to tolerate elevated temperature [175,176]. A major topic in proteome research is the coordinated activation of various pathways governing desiccation tolerance, which is triggered under natural conditions by exposure to low tide [177,178]. Metabolic responses to seaweed pathogens and other aspects of host-pathogen interaction can be explored by comparative proteomic analysis before and after infection [179,180]. In the case of mutation, identification of key proteins is the primary step for the production and/or isolation of stress-tolerant strains of laver. Experimental mutagenesis aims to produce cultivated laver strains capable of efficient growth with resistance to environmental stress factors and has been induced by chemical (ethyl methane sulfonate exposure) [181] or physical (gamma irradiation) treatments [182].

Lipidomic studies have mainly focused on lipid biomarkers, which play an important role in stress-response metabolism, and on the assessment of the nutritional/functional quality of lavers. Elevated habitat temperature as a result of global warming is regarded as a key stress factor for laver, and thus, lipidomic changes under high temperature were examined to determine the acclimation strategies of laver [183,184]. Clear differences in the lipidomic profiles of blade (gametophyte) and conchocelis (sporophyte) stages have identified the main health-promoting lipids characterizing the life cycle stages. This provides reference data for the selection of appropriate life cycle stages for practical uses as food and for functional ingredients and/or biotechnological applications [185].

Since metabolites can determine the nutritional quality of laver and organoleptic characteristics (such as flavor), metabolomic analysis can contribute to the identification of the major metabolites and suggest optimal strategies to produce high-quality products. Metabolomic variations reported in raw, recently harvested laver (changes in glutamine, alanine, aspartate, taurine, and isofloridoside) [186] and in processed laver products during manufacturing (changes in dominant metabolites including amino acids, carboxylic acids, choline metabolites, and sugars) [187] highlight the importance of metabolomic data with respect to the effects of these determinant factors on product quality. Moreover, since metabolite profiles of major edible seaweeds show distinct, species-specific characteristics, metabolomic data specific to laver are also required [188].

## 5. Conclusions

The comprehensive information regarding advances in research reporting health functionality and quality control of laver analyzed in this study demonstrates the need for consistent research studies specializing on laver. Major findings from the analysis of literatures are as follows: (1) nutritional/functional values can be variable according to the type of laver product (raw laver and processed laver products), (2) potential health functionalities linked to the unique substances in laver have been demonstrated by in vitro and in vivo research whereas long-term outcomes of laver consumption on human health should be further examined, (3) intervention methods for chemical/microbial risks should be improved for wider application to manufacturers’ use, (4) omics-technologies have revealed the clues for understanding biological nature linked to product quality of laver. Although previous research studies reported the nutritional/functional values of laver products (raw laver, processed laver products) as useful edible seaweeds, most of those studies highlighted that the accumulation of laver-specific data should be accelerated for the in-depth understanding of the biological nature of laver. The current and future issues described in this study regarding the usefulness of laver are expected to contribute to the balanced progress on both the utilization of lavers as edible seaweeds and the source of the health functionality components.

## Figures and Tables

**Figure 1 marinedrugs-18-00014-f001:**
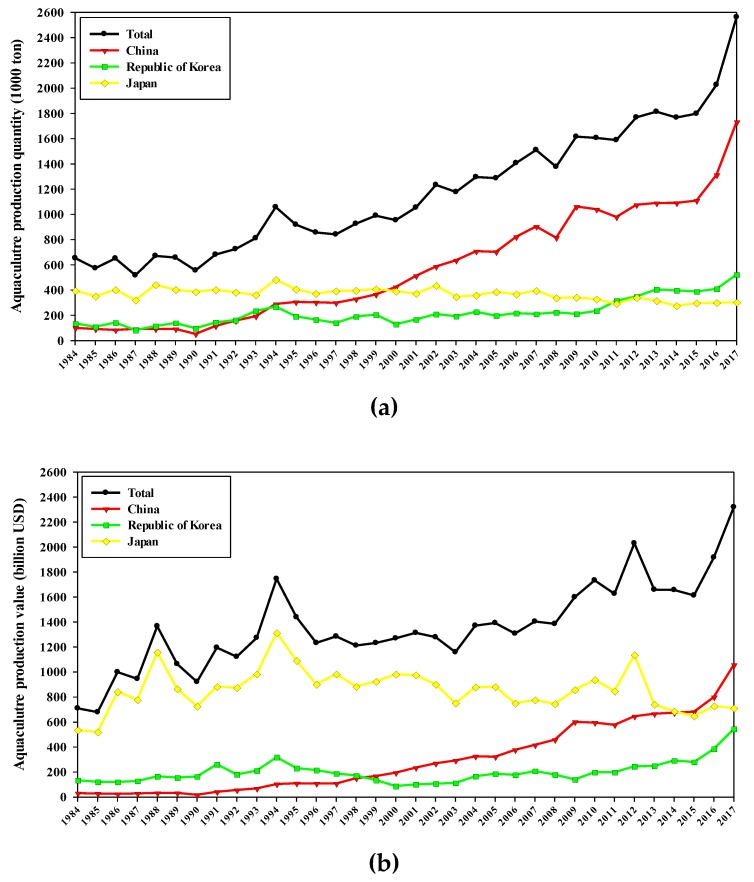
Global aquaculture production of the dominant production regions of the laver: (**a**) production quantity, (**b**) production value. Data (production quantity and value of ‘Laver (Nori)’ for Republic of Korea, ‘Nori nei’ for China, ‘Laver (Nori)’ for Japan) was obtained from FAO’s Fisheries and Aquaculture statistics (FishStatJ) [5].

**Table 1 marinedrugs-18-00014-t001:** Major species of lavers.

Genus	Species ^1^
*Porphyra*	*P. acanthophora, P. columbina, P. dentata, P. dioica, P. fucicola, P. haitanensis,* *P. kanakaensis, P. perforata, P. pseudolinearis, P. purpurea, P. sanjuanensis* *P. seriata, P. tenera, P. umbilicalis, P. vietnamensis, P. yezoensis*
*Pyropia* ^2^	*P. acanthophora, P. columbina, P. dentata, P. fucicola, P. haitanensis,* *P. kanakaensis, P. nitida, P. orbicularis, P. perforata, P. pseudolinearis* *P. seriata, P. tenera, P. vietnamensis, P. yezoensis*

^1^ Species used as the target organisms from research studies cited in this review were summarized. ^2^ This taxonomy was based on the generic revision of laver (*Porphyra* and *Pyropia*) [1].

**Table 2 marinedrugs-18-00014-t002:** Nutritional values of the raw laver.

Product Type	Raw Material (Species)	Nutritional Values from Proximate Analysis (*w/w* %)						Other Nutritional Substances	Reference
		Carbohydrate	Dietary Fiber	Protein	Lipid	Ash	Moisture		
Raw wet laver	*P. yezoensis*	1.2–2.7	- ^1^	3.0–5.0	0.5	3.6–4.3	89.2–90.5	mineral	[19]
	*Porphyra* sp.	-	43.1, 38.9	25.6, 26.0	-	-	-	-	[20]
Raw laver (dry weight) ^2^	*P. vietnamensis*	38.8–60.4	-	12.4–20.5	0.2–2.7	3.9–7.4	13.6–20.7	fatty acids	[21]
	*P. dentata*	45.7–45.9	-	36.2–37.7	0.7–1.0	7.1–8.2	8.6–8.8	mineral, amino acids	[25]
	*P. purpurea*	21.7 ^3^	22.9	33.2	1.0	21.3	-	amino acids, fatty acids, sterol	[24]
	*P. columbina*	-	48.0	24.6	0.3	6.5	12.8	mineral, amino acids, fatty acids, antioxidants, phenolic compounds	[22]
	*P. yezoensis*	51.2–57.9	-	36.2–39.2	2.3–3.1	3.8–7.3	-	mineral, amino acids	[27]
	*P. acanthophora* var. *robusta*	35.5–61.0	-	14.1–18.4	1.7–2.6	4.2–6.8	12.5–21.5	mineral, fatty acids, pigments, vitamin	[26]
	*P. purpurea*	-	-	-	-	-	-	mineral, vitamin	[23]

^1^ Not analyzed or not indicated. ^2^ Dry weight of components of raw laver. ^3^ Non-fibrous.

**Table 3 marinedrugs-18-00014-t003:** Nutritional values of processed laver products.

Category	Product Type ^1^ (Species) ^2^	Nutritional Values from Proximate Analysis (*w/w* %)						Other Nutritional Substances	Reference
		Carbohydrate	Dietary Fiber	Protein	Lipid	Ash	Moisture		
Processed laver products	DL (*Porphyra* spp.)	36.8	31.6	43.0	0.5	10.3	9.4	mineral, amino acids	[8]
	DL (*P. dentata*, *P. seriata*)	47.6	40.4	37.3	0.3	7.6	7.3	fatty acids, pigments, antioxidants	[30]
	DL	43.8–46.2	- ^3^	37.8–40.0	1.5–2.3	8.0–9.0	5.7–7.4	mineral, amino acids, fatty acids, component sugar	[31]
	DL (*P. tenera*)	-	-	36.9	2.3	9.1	3.7	mineral, amino acids	[32]
	DL (*P. haitanensis*)	-	-	32.16	1.96	8.78	6.74	-	
	DL (*P. yezoensis*)	45.4–50.0	-	29.3–35.0	1.8–2.0	8.1–9.9	8.2–9.8	mineral	[19]
	DL	-	-	-	-	-	8.4	phenolic compounds	[33]
	DL	-	-	-	-	-	7.6	-	
	RL	-	-	-	-	-	8.7	-	
	DL	-	-	-	-	-	8.7	-	
	DL (*P. yezoensis*)	-	-	-	-	-	-	vitamin, organic acid, free sugar	[34]
	DL	-	-	-	-	-	-	phenolic compounds, anion, element	[35]
	DL *(P. tenera*, *P. yezoensis* + *P. dentata*, *P. seriata*)	41.7	33.4	38.4	0.3	8.0	11.6	-	[30]
Processed laver products with other seaweeds as optional ingredients	DL combined with green laver (*Ulva* spp.) ^4^	43.7	36.6	35.0	0.8	9.1	11.4	-	[30]

^1^ DL: dried laver; RL: roasted laver. ^2^ Species as the raw material of the product was indicated. ^3^ Not analyzed. ^4^ Nutritional values of processed laver products composed multiple species of optional ingredients.

**Table 4 marinedrugs-18-00014-t004:** Major health functionality of laver products (raw laver and processed laver products).

Health Functionality	Major Components Linked to Health Functionalities	References
Anti-cancer	polysaccharides (dietary fiber, porphyran), phospholipids, sterol, peptide	[17,46,47,48,49,50,51,52,53,54]
Prevention of cardiovascular disease (e.g., hypertension, atherosclerosis, ischemia)	betaine, dietary fiber, taurine, porphyran	[17,55,56,57,58,59,60,61]
Antioxidant effect (e.g., Anti-ageing)	porphyran, glycoprotein, polyphenols, tocopherols, peptide	[62,63,64,65,66,67]
Anti-inflammatory effect and immunomodulation	glycoprotein, porphyran	[64,68,69,70,71,72,73]
Alcohol metabolism	glycoprotein	[74,75]
Prevention of nervous diseases (e.g., Alzheimer’s diseases, methylmalonic acidemias)	taurine, porphyran	[38,76,77]
Prevention of bone disease (e.g., osteoporosis, rheumatoid arthritis)	porphyran, glycoprotein	[64,78]
Anti-diabetes mellitus	phenolic compounds (carotenoids, anthocyanins), polysaccharides (porphyran), peptide	[79,80,81,82]

**Table 5 marinedrugs-18-00014-t005:** Research studies regarding the chemical risk of laver products.

Category	Product Type ^1^	Target	Results (mg/kg or μg /g of dw ^2^)	References
Raw laver	-	arsenic (As)	9.59–34.0	[100]
			22.9–33.8	[101]
			0.22–0.70	[105]
			12.87	[109]
		cadmium (Cd)	0.40–1.21	[100]
			2.83–3.54	[101]
		chromium (Cr)	0.32–0.86	[100]
		copper (Cu)	7.92–16.9	[100]
			1.94–6.94	[101]
		iron (Fe)	290–723	[100]
		lead (Pb)	0.78–1.30	[100]
			<LOD ^3^	[101]
			0.98	[109]
		mercury (Hg)	0.005–0.006	[100]
			0.03	[109]
		nickel (Ni)	0.69–1.04	[100]
			0.74–1.51	[101]
		zinc (Zn)	18.0–57.7	[100]
			21.1–70.1	[101]
Processed laver products	DL	aluminum (Al)	388.6–623.4	[104]
			66–511	[106]
		arsenic (As)	13.5–32.8	[100]
			<LOD–29.850	[102]
			ND ^4^–0.303	[106]
			30.18–39.05	[42]
		cadmium (Cd)	0.69–4.73	[100]
			0.108–3.11	[106]
			0.076–0.318	[42]
			0.501–2.421	[102]
		chromium (Cr)	0.46–0.66	[100]
		copper (Cu)	5.02–8.64	[100]
		iron (Fe)	103–214	[100]
		lead (Pb)	ND–0.86	[100]
			ND	[42]
			<LOD–2.362	[102]
			ND–0.208	[106]
		mercury (Hg)	0.004–0.008	[100]
			0.005–0.009	[42]
			0.002–0.050	[102]
		nickel (Ni)	0.17–1.49	[100]
		zinc (Zn)	27.1–57.7	[100]
	DL, RL	arsenic (As)	2.1–21.6	[107]
	- ^5^	aluminum (Al)	15.50 ^6^	[110]
		arsenic (As)	2.07 ^6^	[110]
		cadmium (Cd)	0.109 ^6^	[110]
		lead (Pb)	0.063 ^6^	[110]
		mercury (Hg)	<LOD	[110]

^1^ DL: dried laver; RL: roasted laver. ^2^ dw: dry weight. ^3^ LOD: limit of detection. ^4^ ND: not detected. ^5^ Specific product type was not indicated in the cited literature. ^6^ Average value.

**Table 6 marinedrugs-18-00014-t006:** Research studies regarding the microbiological risk of laver products.

Category	Target Microorganisms	Product Type ^1^	Results	References ^2^
Processed laver products	Mesophilic bacteria	Standard	4.48 log CFU/g	[113]
		DL	6.5 log CFU/g	[121]
		DL	7.6 log CFU/g	[123]
		RL	7.5 log CFU/g	
		DL	6.9 log CFU/g	[125]
		RL	3.4 log CFU/g	
		SL	4.9 log CFU/g	
		DL	5.6–7.2 log CFU/g	[126]
		RL	3.6 log CFU/g	
		SL	4.3–6.0 log CFU/g	
	Coliforms	Standard	30 MPN ^3^/100 g	[113]
		DL	3.2 MPN/ 100 g	[123]
		RL	3.7 MPN/ 100 g	
		DL	2.1 log CFU/g	[125]
		RL	1.6 log CFU/g	
		SL	1.0 log CFU/g	
		DL	1.9–2.2 log CFU/g	[126]
	Yeast/mold	Standard	2.48 log CFU/g	[113]
		DL	4.3–4.9 log CFU/g	[126]
		RL	2.1 log CFU/g	
		SL	2.1–4.7 log CFU/g	
	*Bacillus cereus*	DL	2.3 log CFU/g	[126]
Raw materials of food products using laver	Mesophilic bacteria	Standard	4.48 log CFU/g	[113]
		DL	8.8 log CFU/g	[127]
		DL	*ca.* 7.0 log CFU/g	[130]
		RL	*ca.* 6.0 log CFU/g	
		DL	5.3 log CFU/g	[131]
	Coliforms	Standard	30 MPN/100 g	[113]
		DL	*ca.* 3.0 log CFU/g	[130]
		RL	*ca.**2*.0 log CFU/g	
		DL	detection rate 6%	[131]
	*B. cereus*	DL	detection rate 12%	[131]
	*Clostridium perfringens*	DL	detection rate 3%	[131]
Work-in-process and end-products from manufacturing plants	Mesophilic bacteria	Standard	4.48 log CFU/g	[113]
		DL	5.6–8.0 log CFU/g	[42]
		DL	4.4–7.8 log CFU/g	[116]
		SL	1.3–5.9 log CFU/g	
		DL	4.7–4.8 log CFU/g	[132]
		SL	ND ^4^–1.0 log CFU/g	
		DL	3.4–3.6 log CFU/g	[133]
		SL	1.4–2.8 log CFU/g	
	Coliforms	Standard	30 MPN/100 g	[113]
		DL	54–27,600 MPN/100 g	[42]

^1^ DL: dried laver; RL: roasted laver; Standard: permitted values standardized by China which has strict regulations (GB 2733-2005) were indicated as reference data [113]. ^2^ This table is adapted and modified from [91].^3^ MPN: Most probable number.^4^ ND: Not detectable.

**Table 7 marinedrugs-18-00014-t007:** Intervention methods for microbial potential risks of processed laver products.

Target Product	Treatment Methods	Target Microorganisms	Treatment Conditions	Microbial Reduction (log CFU/g)	References
Raw harvested laver	NaOCl + ultrasound	*Escherichia coli* *Bacillus cereus*	200 ppm, 60 min	2.6 3.2	[134]
*Kimbab*	Gamma irradiation	Mesophilic bacteria	1–3 kGy, 24 h	1.0–2.0	[130]
*Kimbab*	Gamma irradiation	*Escherichia coli* *Salmonella* Typhimurium *Staphylococcus aureus* *Listeria monocytogenes*	1–3 kGy, 24 h	1.3–ND ^1^ 2.3–ND 3.4–ND 2.7–ND	[120]
DL ^2^	UV	Mesophilic bacteria	20 W, 20 min	1.0	[123]
DL	Gamma irradiation	*Escherichia coli* *Salmonella* Typhimurium *Staphylococcus aureus* *Listeria ivanoviis*	1–3 kGy, 24 h	2.7–ND 1.7–ND 2.0–ND 1.6–ND	[127]
DL	Corona discharge plasma	Mesophilic bacteria	3312 rpm, 58Hz, 20 min	2.0	[135]
DL	Low–pressure air plasma	Mesophilic bacteria	20 min	1.5–2.0	
DL	e-beam	Mesophilic bacteria	4 kGy	1.4	[136]
DL	Heat-assisted e-beam irradiation	Mesophilic bacteria	1.8–3.0 kGy, 154–170 °C, 10–18 s	> 2.0 ^3^	[139]
DL	Heat-assisted e-beam irradiation	Coliform	4 kGy, dose rate as 2.1 kGy/h	> 1.5 ^4^	[137]
DL	Heat-assisted low-dosee-beam irradiation	Coliform	1 kGy, dose rate as 2.1 kGy/hg	> 1.4 ^5^	[138]

^1^ ND: not detected. ^2^ DL: dried laver. ^3^ Microbial reduction was calculated from the control group of this study (no treatment of e-beam irradiation) as heating 160 °C for 14 sec without e-beam irradiation. ^4^ Initial population level of coliform was 2.5 log CFU/g and the irradiation reduced coliform to undetectable levels with the detection limit as 1 log CFU/g. ^5^ Initial population level of coliform was 2.4 log CFU/g and the irradiation reduced coliform to undetectable levels with the detection limit as 1 log CFU/g.

**Table 8 marinedrugs-18-00014-t008:** Omics-based studies linked to the health functionality and the processing of lavers.

Omics Technology	Topic	Species	Major Findings	References
Genome	Whole genome sequencing and genomic feature	*P. umbilicalis*	- Genome governing nutritional/functional values linked to the growth and survival strategy of laver under stressful condition of natural habitat (intertidal zone)	[147]
		*P. yezoensis*	- First report on the genome sequence of nuclear ribosomal DNA (nrDNA) cistron	[149]
		*P. yezoensis*	- Genome sequence and annotated functional genes from *P. yezoensis* - Identification of photosynthesis system and key genes governing color of laver	[154]
	Genome-wide identification of functional genes	*P. yezoensis*	- Gene structure associated with mitogen-activated protein kinases from *P. yezoensis* (*PyMAPKs*)	[156]
	Comparative genomics	*P. perforata* *P. sanjuanensis* *P. fucicola* *P. kanakaensis*	- Reliable analytical method for the genomes of laver by the destructive sampling of type specimen	[146]
		*P. nitida*	- Recognition of new red algal species	[148]
		*P. haitanensis*	- Supportive data for the phylogenic differences between *Pyropia* from *Porphyra*	[150]
		*P. yezoensis*	- Supportive data for the phylogenic differences between *Pyropia* from *Porphyra*	[151]
		*P. yezoensis*	- Different genomic structure of strains according to the regions of cultivars (Korea and China)	[152]
		*P. haitanensis*, *P. yezoensis*	- Biodiversity and distinct phylogenies of laver compared with other red algae	[153]
Transcriptome	Analytical techniques	*P. haitanensis*	- Selection of housekeeping gene mostly adequate for the designation of internal control based on the stability under abiotic stresses	[157]
	Unique life cycle	*P. yezoensis*	-Transition observed in the life cycle with apospory	[158]
		*P. umbilicalis,* *P. purpurea*	- Evolutionary analysis for the growth and development of laver	[159]
		*P. haitanensis*	-Transcriptomic profile under different physiological conditions -Role of cSSR markers linked to the differences in the gene expressions among lifecycle stages of laver	[160]
		*P. pseudolinearis*	- Impact of ethylene precursor treatment to the regulation of gene expression governing reproduction	[161]
	Stress response	*P. yezoensis*	- Stress response of PyMAPK gene family	[156]
		*P. yezoensis*	- Identification of key response genes expressed under various abiotic stresses	[162]
		*P. haitanensis*	- Role of heat shock proteins against the abiotic stresses	[163]
		*P. tenera*	- Distinct transcriptional characteristics of gametophyte thalli by high-temperature stresses	[164]
		*P. yezoensis*	- Transcriptomic profiles in response to stresses associated with temperature	[165]
		*P. haitanensis*	- Identification of key response genes expressed under thermal stresses - Mechanisms on the adaptation of high-temperature tolerant strain	[166]
		*P. columbina*	- Identification of mechanisms on resistance and key response genes expressed under stresses from desiccation-hydration cycles in natural habitat	[167]
		*P. tenera*	- Identification of mechanisms on resistance and key response genes expressed under desiccation	[168]
		*P. haitanensis*	- Identification of mechanisms on resistance and maintenance of homeostasis under stresses from hypersaline conditions	[169]
	Biosynthesis	*P. yezoensis*	- Role of glycine-betaine (GB) capable of maintenance of osmotic balance in response to desiccation stresses - Identification of major enzymes involved in the biosynthesis of GB	[189]
Microbiome	Diversity in the microbiota	*P. umbilicalis*	- Seasonal variation to the microbial community in laver - Identification of bacterial groups which are expected to contribute to the evolution and/or function of laver	[155]
		*P. yezoensis*	- Seasonal variation and the effects of the yellow spot disease outbreaks to the microbial community in the seawater of laver seedling pools - Identification of disease-associated bacteria	[174]
	Analytical techniques	*P. umbilicalis*	- Microbial communities affected by the sampling position of laver and the stabilization techniques applied for the microbiome analysis	[171]
	Influencing factor (Red rot disease)	*P. yezoensis*	- Alterations of bacterial community by red dot disease - Close association between health status of algal host (uninfected or infected) and bacterial community	[190]
Proteome	Mechanism of stress-tolerance	*P. haitanensis*	- Investigation on the key metabolisms elucidating the mechanisms of resistance to high-temperature	[175,176]
		*P. orbicularis*	- Investigation on the key metabolisms elucidating the mechanisms of resistance to desiccation	[177,178]
		*P. haitanensis*		
	Mechanism of infection resistance	*P. yezoensis*	- Investigation on the pathogen-responsive proteins elucidating the mechanisms of responses against the infection	[179]
	Identification of key functional protein	*P. yezoensis*	- Identification of major protein [*Pyropia yezoensis* aldehyde dehydognease (PyALDH)] which contributes to the resistance of laver against oxidative stress	[180]
	Mutation of laver strain	*P. yezoensis*	- Induction of high-growth-rate mutation by the exposure to ethyl methane sulfonate - Comparative analysis for the proteome of mutated strain with wild-type strain with the perspective to the enhanced growth	[181]
		*P. yezoensis*	- Induction of thermo-tolerance mutation by the exposure to gamma-irradiation - Isolation of protein from thermo-tolerant mutant which contributes to the resistance against elevated temperature	[182,191]
Lipidome	Lipidomic variations	*P. haitanensis*	- Identification of lipid biomarkers distinctly expressed under elevated temperatures	[184]
		*P. dioica*	- Differences in composition of major lipid molecular species according to the life cycle stages between the blade and conchocelis	[185]
Metabolome	Metabolomic variations	*P. haitanensis*	- Changes in the nutrient composition according to the harvest time	[186]
		*P. yezoensis*	- Changes in the nutrient composition which can determine the taste of laver by the food processing steps not only for seasoning but also washing, cutting, and roasting	[187]
	Metabolite profile	*P. pseudolinearis*	- Distinctive characteristics of metabolites among species of edible seaweeds (brown, red, and green algae) and sorbitol as the major sugar metabolite in laver	[188]

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
