# Peer review of "Health Functionality and Quality Control of Laver (Porphyra, Pyropia): Current Issues and Future Perspectives as an Edible Seaweed"

_marinedrugs, 2019, doi:10.3390/md18010014_

Round 1

Reviewer 1 Report

BRIEF SUMMARY

This review compiles information about: type of foods products containing lavers; major components linked to health benefits, when considering lavers as functional food; laver food safety; findings about technological advances in food processing to minimize the microbiological and chemical risks; and research findings about omics-technology to identify the health-promoting properties of laver.

STRENGTH: The topic of this review is interesting and timely, as it focuses on a future alternative food source in the world (not only Asia).

WEAKNESS: The review compiles information about research findings, but lacks a search for patents and actual commercial applications. Section 4.2 is not well related to the previous text; it seems to constitute another paper per se. The Conclusions section is too vague, and it does not report comments highlighting major conclusions reached from the compilation of information in the review. Some statements are made without support.

SPECIFIC COMMENTS

Line 2: The title is too vague, because the article focuses on laver as a food product.

Line 34: “The growth of global seaweed aquaculture as a source of pharmaceutical and biomaterials has also contributed to the expansion of the laver industry.”

I suggest giving examples of some products and companies available in the market. At least, give examples of the larger ones that contributed to the expansion of laver industry.

Line 38: “Commercial production and export…. 99.87% of total world production in 2017”. Some of the most important products should be named as examples.

Line 46: “ Among production … of substances beneficial to health (Table 1).”

What type of food and what kind of substances are the authors referring to?

Line 54: Add two columns in Table 1 – one specifying the type of edible seaweed products, and the other indicating the type of functional substances. The same with regard to Table 2.

Line 62: Table 2 not Table 1 - confirm.

Line 73: I suggested some patent search search.

Line 84: Explain the reasoninh better, because 90+20 =110%?

Line 149: “However, … does not always implicate the processing steps as casual factors.” Why? Explain it better. What are the scientific basis for this sentence?

Line 167: Major health functionally of the lavers as what? Raw food, processed food (roasted, dried, etc..), supplement? This is important to be explain because the authors mentioned previously in the text that active components in laver might be lost during food treatment.

Line 240: PTWI values should be added to Table 5, for comparison purposes.

Line 245: Data in Table 5 are divided according to the references – but they should be divided according to target. It will give a better presentation and allow comparison of results.

Line 245: Why are some of the results for the same target so different. Add some comment on that. Were they harvested from a different region, at different cultivar stages, or used a different protocol of measurement?

Line 267: The maximum permitted values for microorganisms, according to China regulations, should be inserted in Table 6 to facilitate analysis of the data. In this table, the information is listed by reference instead by target microorganisms (these should be grouped). Why are some of the data related to coliforms given by 100 g and not by 1 g?

Line 337: “Intervention methods for microbial potential risks … “ (Table 7) are results from research findings. What about patents? Why are there a limitation in manufacturer’s use? More information and discussion about this should be given.

Line 352: I do not understand the connection of this section with the text previously discussed. Make it more clear.

Line 434: Omics technology order is good. Once again, the information is listed by reference instead of (in this case) by species name. The major findings are to be compared but, in the way they are given, they are spread over all the table.

Author Response

Response to Reviewer 1 Comments

Point 1: BRIEF SUMMARY and STRENGTH/WEAKNESS

BRIEF SUMMARY: This review compiles information about: type of foods products containing lavers; major components linked to health benefits, when considering lavers as functional food; laver food safety; findings about technological advances in food processing to minimize the microbiological and chemical risks; and research findings about omics-technology to identify the health-promoting properties of laver.

STRENGTH: The topic of this review is interesting and timely, as it focuses on a future alternative food source in the world (not only Asia).

WEAKNESS: The review compiles information about research findings, but lacks a search for patents and actual commercial applications. Section 4.2 is not well related to the previous text; it seems to constitute another paper per se. The Conclusions section is too vague, and it does not report comments highlighting major conclusions reached from the compilation of information in the review. Some statements are made without support.

Response 1: Thank you for helpful comment. We revised the manuscript according to your evaluation for our manuscript with the perspective to the ‘WEAKNESS’: most of comments were applied by responses to specific comments (From Point 2 to Point 18 as shown below), whereas conclusion section was revised to highlight major conclusions from information in this review.

Point 2: Line 2: The title is too vague, because the article focuses on laver as a food product.

Response 2: We suggested new title for this review following your comment.
à Health functionality and quality control of laver (Porphyra, Pyropia): current issues and future perspectives as an edible seaweed.

Point 3: Line 34: “The growth of global seaweed aquaculture as a source of pharmaceutical and biomaterials has also contributed to the expansion of the laver industry.”. I suggest giving examples of some products and companies available in the market. At least, give examples of the larger ones that contributed to the expansion of laver industry.

Response 3: We revised the sentence by adding examples of representative companies for seaweed products (Line 35 in the revised manuscript).

Point 4: Line 38: “Commercial production and export…. 99.87% of total world production in 2017”. Some of the most important products should be named as examples.

Response 4: Since the data regarding the world laver production suggested in the manuscript are based on the overall statistics for laver products regardless of the product type, we added the information about products as “(raw laver and processed laver products)” in Line 40 of the revised manuscript.

Point 5: Line 46: “ Among production … of substances beneficial to health (Table 1).” What type of food and what kind of substances are the authors referring to?

Response 5: Laver species provided in Table 1 are target organisms used in the researches cited in this review. Thus, we revised this sentence to emphasize the source of information as researches on the food production (raw and/or processed laver) and/or substances potentially beneficial to health (Line 47–50 in the revised manuscript). We also added the footnote under Table 1 for the explanation of this issue (Line 56 in the revised manuscript).

Point 6: Line 54: Add two columns in Table 1 – one specifying the type of edible seaweed products, and the other indicating the type of functional substances. The same with regard to Table 2.

Response 6: Since species indicated in Table 1 are selected according to the summary of target species from researches cited in this review (as described in the ‘Response 5’), specific type of edible seaweed products or functional substances cannot be specified for each species. To avoid misunderstanding, we revised the title of Table 1 (Major species of lavers.) and added the footnote under Table 1 for the explanation of this issue (as described in the ‘Response 5’).

Point 7: Line 62: Table 2 not Table 1 - confirm.

Response 7: To avoid confusing readers, we deleted the parentheses indicating ‘Table 1’ from Line 63 in the revised manuscript. We intended to indicate ‘algal species’ (Line 63 in the revised manuscript) for ‘(Table 1)’.

Point 8: Line 73: I suggested some patent search search.

Response 8: We added the name of international database (WIPO IP Portal) used for searching patents cited in this review.

Point 9: Line 84: Explain the reasoninh better, because 90+20 =110%?

Response 9: To prevent confusing readers, we deleted the sentence indicating water content by the comparison between wet weight and dry weight (Line 87-89 in the revised manuscript).

Point 10: Line 149: “However, … does not always implicate the processing steps as casual factors.” Why? Explain it better. What are the scientific basis for this sentence?

Response 10: This statement was based on the next sentence (Line 154-156 in the revised manuscript). To explain the scientific basis for the statement, we connected two sentences (Line 152-156 in the revised manuscript).

Point 11: Line 167: Major health functionally of the lavers as what? Raw food, processed food (roasted, dried, etc..), supplement? This is important to be explain because the authors mentioned previously in the text that active components in laver might be lost during food treatment.

Response 11: We revised the title of Table 4 according to your comment [Major health functionality of laver products (raw laver and processed laver products)]. Major components linked to health functionality shown in Table 4 are inherent substances in raw laver. However, although those components are likely to be lost partly during food treatment, most components can remain in processed laver products as well.

Point 12: Line 240: PTWI values should be added to Table 5, for comparison purposes.

Response 12: We agree with the necessity for providing information regarding PTWI for heavy metals as your comment. Health based guidance values for tolerable intake [PTWI, provisional tolerable monthly intake (PTMI), provisional maximum tolerable daily intake (PMTDI)] for major heavy metals detected from laver were added in the manuscript (Line 247-250 in the revised manuscript) based on the database provided by JECFA/WHO. In the case of Table 5, results are indicated as contamination level in laver (not as level of expected intake) and thus we think that the provision of PTWI values in Table 5 may confuse readers.

Point 13: Line 245: Data in Table 5 are divided according to the references – but they should be divided according to target. It will give a better presentation and allow comparison of results.

Response 13: Table 5 was revised by the division of data according to the target heavy metal.

Point 14: Line 245: Why are some of the results for the same target so different. Add some comment on that. Were they harvested from a different region, at different cultivar stages, or used a different protocol of measurement?

Response 14: Contents of heavy metals in laver are varied according to the several factors including laver cultivar, species, regions for the collection of laver samples, time (seasonal variation), and processing conditions. We emphasized this issue by the statements (Line 242-244 in the revised manuscript) in the following sentence for the indication of Table 5.

Point 15: Line 267: The maximum permitted values for microorganisms, according to China regulations, should be inserted in Table 6 to facilitate analysis of the data. In this table, the information is listed by reference instead by target microorganisms (these should be grouped). Why are some of the data related to coliforms given by 100 g and not by 1 g?

Response 15: Maximum permitted values for each target microorganism were inserted in Table 6. Information in Table 6 were also grouped by target microorganisms instead of reference number. In the case of reports regarding coliforms in samples given by 100 g, those studies adopted the analytical methods for the results of contamination level as most probable number (MPN) method and the minimal unit was MPN/100 g indicated in those studies.

Point 16: Line 337: “Intervention methods for microbial potential risks … “ (Table 7) are results from research findings. What about patents? Why are there a limitation in manufacturer’s use? More information and discussion about this should be given.

Response 16: Representative patents specific for laver processing are now cited in the statements regarding current issues on intervention methods against microbial risks (Line 347-349 in the revised manuscript). We also stated the reason for major limitation in the application of those intervention methods to wide use for manufacturers (Line 349-355 in the revised manuscript). We really appreciate that your comment is very helpful and inspirational for me. We want to respectively ask your indulgence for this revision. Thank you.

Point 17: Line 352: I do not understand the connection of this section with the text previously discussed. Make it more clear.

Response 17: We added the statements for the explanation of the connection of this section with the topic of this review (i.e. Health functionality and quality control of laver) (Line 367-374 in the revised manuscript).

Point 18: Line 434: Omics technology order is good. Once again, the information is listed by reference instead of (in this case) by species name. The major findings are to be compared but, in the way they are given, they are spread over all the table.

Response 18: Information in Table 8 was re-ordered according to the reference number instead of species name, and was grouped with major topic for each omics technology.

Reviewer 2 Report

The current manuscript is a timely review of one of the major dietary macroalgae consumed around the world. The following suggestions are aimed at improving the critical content and overall appropriateness and clarity of the manuscript.:

Line 58-59 – please reword this sentence. Laver cannot be considered as a “health benefit”. Is this referring to a “positive item/group to include in the diet” or similar? Please reword for clarity and sense. Line 68 – “synthesizes” appears to be the wrong word. Consider revising for clarity to “evaluates”, “considers” or something similar. Section 2.1 (and throughout) – it’s incorrect to describe compositional data on lavers and being “health benefits”. Even in the case of (essential) vitamins and minerals, overall dietary habit defines total intake of each vitamin and mineral. Consuming more of each nutrient could therefore possibly be expected to improve health (in the case of inadequate intake), have no effect on health (if total intake is already adequate) or even result in negative health consequences (if intake is too high for e.g. fat-soluble vitamins and some minerals). I suggest carefully checking through the manuscript and being careful to refer to compositional data in reference to being sources of essential, non-essential and toxic nutrients for scientific appropriateness. “Positive” nutritional profiles could be considered as providing dietary sources of nutrients X, Y and Z. Line 115-166 – see comment above. This sentence end with the “…potential for health benefits arising…” or similar. Section 2.2 – it may be worth highlighting that the process of drying greatly improves shelf-life of laver and other seaweeds, thereby ensuring they can be accessed by a wider number of individuals as a food source. Section 3 – this section needs to be expanded greatly. At the moment, the authors are only reporting what previous studies showed. It is also important to consider the quality and strength of evidence in relation to long-term outcomes on human health. Separation of evidence on laver consumption and consumption of phenolics/nutrients that might exist in laver are also of major importance to provide the reader with critical evaluation of the existing evidence base. This should importantly also feed into highlighting the need for future work in the next section (“4. Future perspectives on technical advances in laver utilization”). Line 437 – update “the researches” to “research” or similar. Line 439-442 – I’m not sure that the authors have managed to effectively highlighted that laver is compositionally distinct from other seaweeds by the preceding information. Please consider revision of this statement for appropriateness.

Author Response

Response to Reviewer 2 Comments

Point 1: The current manuscript is a timely review of one of the major dietary macroalgae consumed around the world. The following suggestions are aimed at improving the critical content and overall appropriateness and clarity of the manuscript.:

Response 1: Thank you for providing us helpful comments. We checked the manuscript with the perspective of your comments (from Point 2 to Point 9) and revised the previous version of the manuscript to improve the overall quality.

Point 2: Line 58-59 – please reword this sentence. Laver cannot be considered as a “health benefit”. Is this referring to a “positive item/group to include in the diet” or similar? Please reword for clarity and sense.

Response 2: We replaced the expressions “health benefit” to “source of health functionality substances” (Line 60 in the revised manuscript). Thank you for critically pointing out the issue to be improved.

Point 3: Line 68 – “synthesizes” appears to be the wrong word. Consider revising for clarity to “evaluates”, “considers” or something similar.

Response 3: We replaced the word “synthesizes” to “evaluates” to follow your comment (Line 69 in the revised manuscript).

Point 4: Section 2.1 (and throughout) – it’s incorrect to describe compositional data on lavers and being “health benefits”. Even in the case of (essential) vitamins and minerals, overall dietary habit defines total intake of each vitamin and mineral. Consuming more of each nutrient could therefore possibly be expected to improve health (in the case of inadequate intake), have no effect on health (if total intake is already adequate) or even result in negative health consequences (if intake is too high for e.g. fat-soluble vitamins and some minerals). I suggest carefully checking through the manuscript and being careful to refer to compositional data in reference to being sources of essential, non-essential and toxic nutrients for scientific appropriateness. “Positive” nutritional profiles could be considered as providing dietary sources of nutrients X, Y and Z.

Response 4: Thank you for this significant comment. We agree that your opinion should be applied to overall contents of Section 2, and added the statements for this issue at the top of Section 2 based on the comment (Line 80-83 in the revised manuscript).

Point 5: Line 115-166 – see comment above. This sentence end with the “…potential for health benefits arising…” or similar.

Response 5: Following your comment, we revise the sentences referring the health benefits. (Line 118-165 in the revised manuscript).

Point 6: Section 2.2 – it may be worth highlighting that the process of drying greatly improves shelf-life of laver and other seaweeds, thereby ensuring they can be accessed by a wider number of individuals as a food source.

Response 6: Thank you for suggesting the improvement of manuscript. We stated the advantages for the drying process in Section 2.2 (Line 158-161 in the revised manuscript) following your comment. 

Point 7: Section 3 – this section needs to be expanded greatly. At the moment, the authors are only reporting what previous studies showed. It is also important to consider the quality and strength of evidence in relation to long-term outcomes on human health. Separation of evidence on laver consumption and consumption of phenolics/nutrients that might exist in laver are also of major importance to provide the reader with critical evaluation of the existing evidence base. This should importantly also feed into highlighting the need for future work in the next section (“4. Future perspectives on technical advances in laver utilization”).

Response 7: According to your comments, we revised the Section 3 with the perspectives to 1) importance to consider the quality and strength of evidence in relation to long-term outcomes on human health and emphasizing the necessity for investigating long-term outcomes of laver consumption on human health as a future research, 2) separation of evidence on laver consumption and consumption of phenolics/nutrients that might exist in laver. Firstly, we added the statement regarding the limitation of references currently cited in the manuscript and the necessity of researches on long-term outcomes on human health in both Section 3 (Line 171-174 in the revised manuscript) and Section 4 (Line 370-373 in the revised manuscript). Secondly, for each substance described in this section (porphyran, VitB12, and taurine), we provided additional explanation for the separation of evidences on laver consumption and consumption of health functional substances extracted from laver (Overall contents in the paragraphs from Line 180-213 in the revised manuscript). We want to say that we acknowledge your valuable comments.

Point 8: Line 437 – update “the researches” to “research” or similar.

Response 8: The word “the researches” is now updated to “research” (Line 456 in the revised manuscript).

Point 9: Line 439-442 – I’m not sure that the authors have managed to effectively highlighted that laver is compositionally distinct from other seaweeds by the preceding information. Please consider revision of this statement for appropriateness.

Response 9: We agree with your opinion for the statement of distinct characteristics of laver compared with other seaweeds. To avoid misunderstanding, we revised the sentence as follows: ‘Although previous researches reported the nutritional/functional values of laver products (raw laver, processed laver products) as useful edible seaweeds, most of those studies also highlighted that the accumulation of laver-specific data should be accelerated for the in-depth understanding of biological nature of laver.’ (Line 465-468 in the revised manuscript).

Round 2

Reviewer 1 Report

Line 39: give some names names of commercial products; remove raw laer and processed laver products.

Line 50: give some examples of substances potentially beneficial to health

Table 5: due to large variability of data it is difficult to understand which results are for the target; I suggest to put the target on first line of the results.

Author Response

Point 1: Line 39: give some names names of commercial products; remove raw laer and processed laver products.

Response 1: We deleted “raw laver and processed laver products” and rather added representative examples for the names of commercial products (Line 38-41 in the revised manuscript).

Point 2: Line 50: give some examples of substances potentially beneficial to health

Response 2: Major substances unique in laver (as described in Section 3; porphyran, taurine, vitamins) are suggested as representative examples (Line 51 in the revised manuscript).

Point 3: Table 5: due to large variability of data it is difficult to understand which results are for the target; I suggest to put the target on first line of the results.

Response 3: In the case of ‘Tables’ in the initial version of the manuscript, we put the target on first line of the results for the visibility, however, the target was then aligned at the centre after the editorial revision by the journal ‘Marine drugs’. Since we cannot put the target on first line of the results due to the mandatory template, rather we added the blank line between targets to separate the data within the same target. This revision was also applied to Table 6 and Table 8 as well. Thank you for the helpful comments.

Reviewer 2 Report

I would like to thank the authors for improving their manuscript significantly. I have a few more minor comments aimed at helping improve the precision of wording around the updated sections.

Lines 172-173, L347 and headings of Table 5 and 6 - all uses of the word "researches" should be changed to "research" or "research studies". Please also check the rest of the manuscript for this grammatical issue. Line 172 - "health functional effects" does not quite sound correct. Should this be "putative health effects" or similar? Line 247 - remove "health-based". I think these guidelines are probably more risk-based.

Author Response

Point 1: Lines 172-173, L347 and headings of Table 5 and 6 - all uses of the word "researches" should be changed to "research" or "research studies". Please also check the rest of the manuscript for this grammatical issue.

Response 1: All terms “researches” used in this manuscript were replaced as “research studies” or “research”.

Point 2: Line 172 - "health functional effects" does not quite sound correct. Should this be "putative health effects" or similar?

Response 2: We replaced the expression “health functional effects” to “putative health effects” by following your comment. Thank you for the suggestion of adequate terms and the expression.

Point 3: Line 247 - remove "health-based". I think these guidelines are probably more risk-based.

Response 3: To follow your comment, we removed the terms “health-based”. We really appreciate that your comments are very helpful for us.